# Effect of Step Size on the Formability of Al/Cu Bimetallic Sheets in Single Point Incremental Sheet Forming

**DOI:** 10.3390/ma16010367

**Published:** 2022-12-30

**Authors:** Krzysztof Żaba, Sandra Puchlerska, Łukasz Kuczek, Tomasz Trzepieciński, Piotr Maj

**Affiliations:** 1Faculty of Non-Ferrous Metals, AGH University of Science and Technology, Al. Mickiewicza 30, 30-059 Krakow, Poland; 2Department of Manufacturing and Production Engineering, Faculty of Mechanical Engineering and Aeronautics, Rzeszow University of Technology, Al. Powst. Warszawy 8, 35-959 Rzeszów, Poland; 3Faculty of Materials Science and Engineering, Warsaw University of Technology, Woloska 141, 02-507 Warsaw, Poland

**Keywords:** Incremental Sheet Forming, bimetallic Al/Cu sheets, maximum forming angle, mechanical properties, roughness, hardness, ANOVA

## Abstract

Single Point Incremental Forming (SPIF) is an unconventional forming process that is suitable for prototype production and small lot production due to the economical tooling cost, short lead time, and the ability to create symmetrical and asymmetrical complex geometries without the use of expensive dies. This article presents the effect of the step size Δz of a forming tool made of 145Cr6 tool steel on the formability and maximum forming angle, mechanical properties, hardness, surface roughness, microstructure and texture of bimetallic Al/Cu sheets. Experiments were conducted at a constant rotational speed and feed rate, with the use of rapeseed oil as a lubricant. The tests were carried out with the use of a forming tool on both sides of the bimetallic sheet. The shape and dimensions of the formed elements are determined by non-contact optical 3D scanning. It has been proved that an increase in the step size Δz affects the deterioration of the surface quality of the specimens (an increase in the Ra parameter from 0.2 μm to approximately 3 μm for the step size of 1.2 mm), while a small step size down Δz favours the geometric stability of the samples. With increasing step size (at Δx = Δy = const.), the drawpiece wall continually thinned until the material fractured. Based on the results, it was shown that increasing the step size Δz over 1.1 mm causes cracking of the drawpieces. Furthermore, greater thinning of the Al/Cu sheet was observed in the range of step size Δz between 0.7 and 1.0 mm for aluminum side and step size Δz ≤ 0.6 mm and Δz ≥ 1.1 mm for copper side. It was also found that the mechanical properties of the bimetal sheet decreased as a result of incremental forming. The greatest decrease in strength and ductility was recorded for a pitch of 1.2 mm. Strength decreased from 230 MPa (for sheet in initial state) to approximately 80 MPa, elongation from 12% to approximately 8.5%, and hardness from 120 HV10 for Cu and 60 HV10 for Al to approximately 30 HV10 for both layers.

## 1. Introduction

Conventional plastic forming processes are used in many industrial sectors. The most common methods for sheet metal forming are stamping, deep drawing, bending, flow forming and shear spinning, in which highly specialised pieces of equipment such as forming dies, punches and presses are used. Consequently, the tools for conventional processes are very expensive and require long lead times. Moreover, when multi-stage processes are used in order to produce a finished product, it is necessary to use many sets of tools, which further increases the costs and implementation time [1]. Therefore, non-conventional plastic forming processes are becoming more and more popular, allowing costs and total time of implementation to be reduced, while the required quality is maintained as identified by the shape, dimensions, properties and surface finish of the products. One such process is Incremental Sheet Forming (ISF), which was patented by Leszak in 1967 [2]. In this process, a tool with a spherical tip mounted on the head of the CNC machine performs circular movements while plunging into the material along the vertical *Z* axis. In its basic variety, Single Point Incremental Forming (SPIF), the edge of the workpiece is fixed in a special frame that prevents the material from flowing freely along all the axes. The rotational tool performs sliding movements according to a given trajectory, carrying out successive stages of incremental forming [3]. The drawpiece is mostly formed as a result of the thinning of the drawpiece wall in the presence of biaxial tensile stresses.

In addition to the basic version of ISF, several variations of the SPIF process are known, including Two Point Incremental Forming (TPIF), Partial Die Incremental Forming (PDIF), Full Die Incremental Forming (FDIF) [4] and Asymmetric Single Point Incremental Forming (ASPIF) [5]. For complex geometries, the SPIF process is more flexible and economical due to its higher formability and shorter lead times compared to conventional forming processes. Therefore, SPIF processes are very suitable for use in the aerospace and biomedical sectors for rapid prototyping and small-lot production [5,6].

Despite the fact that Leszak’s patent is over 50 years old, an increased intensity of research into the ISF process can be observed in the recent decade. The influence of the process parameters on limiting the formability of the sheet metal and the final quality of the products is still not fully understood. Researchers largely undertake analysis of the mutual relations of SPIF process parameters, mainly by performing experiments on single-layer metallic sheets [7]. Usually, the investigations are concerned with the analysis of the influence of tool rotational speed, feed rate, tool trajectory [8], type of material and punch geometry [9] on workpiece formability and optimisation of the SPIF process [10]. In addition to experimental work, numerical simulations of the ISF process are also carried out [11,12]. Attempts have also been made to increase the efficiency of the ISF process through the use of robots [13], hybrid forming [14] and an extremely large increase in tool rotational speed and feed rate [15,16].

One of the factors that was investigated is the step size Δz of the forming tool. In their research, Ham and Jeswiet [17] showed that, in the case of single-layer materials, the step size Δz has no direct impact on the formability of the material. It affects the roughness of its outer and inner surfaces and creates characteristic tool marks. In addition, a smaller step size significantly increases the forming time. Fontanari et al. [18] formed metallic sheets with a step size of 0.5 mm and 0.71 mm. They concluded that the increase in the step size did not change the surface quality of the product.

Currently, the materials that are most frequently processed with the use of ISF are steel sheets [19], Cu and Cu alloys [20], Al and Al alloys [21], Mg alloys [22] and Ti alloys [23]. Attempts have also been made to deform polymer sheets [24]. An interesting topic is the forming of bimetallic sheets, i.e., layered composite materials obtained by joining two or more materials with different properties. Laminated metallic materials are joined by a variety of techniques, including diffusion welding.

The great advantage of bimetallic composites is that they have different properties on the two sides of the sheet [25]. These materials have great potential and therefore the work on developing new techniques for forming them is fully justified. However, the results of research to date on ISF of bimetallic materials are limited to a few references. Gheysarian and Honarpisheh [26] investigated the effects of process parameters on the incremental forming of explosive-welded Al/Cu bimetals. The results of the mathematical model show that the method could provide a suitable description of the operation and corresponds with the experimental results. Liu and Li [27] studied the deformation behaviors of roll-bonded Cu/Al composite sheets in SPIF through numerical as well as empirical approaches and experimental work. It was found that thickness variation and surface roughness in different layer arrangements, in terms of various process parameters, follow the similar trends to single-layer sheets. It was also revealed that there is little difference in the maximum height of the assessed profile of tool-sheet contact surface between two kinds of layer arrangements in SPIF of the truncated pyramids. Honarpisheh et al. [28] carried out an experimental and numerical investigations on single-point incremental forming of explosive bonded clad sheets. They used ANOVA method for evaluation of the interaction and main effect of the process parameters. They concluded that the main effect plots revealed that the higher level of step down with lower level of tool diameter and rotational speed provide higher fracture depth. Sakhtemanian et al. [29] applied the ISF method using ultrasonic vibrations to form steel/Ti bimetallic sheets. Microscopic observation of specimens showed equiaxed fine grains along the primary grain boundaries in which their volume in the structure was increased by increasing the vertical step down. Sakhtemanian et al. [30] carried out experimental and numerical investigations on explosively welded low carbon steel/commercially pure titanium bimetallic sheet. The results showed that by increasing the step size Δz, hardness, tensile properties and wall thickness of the specimens increased and the surface quality decreased. Rahmatabadi et al. [31] investigated the effect of ultrasonic vibrations on the formability of the annealed Al5052/MgAZ31B composite. The results of the biaxial stretch-forming test revealed that the cold rolling process can significantly enhance microhardness and tensile strength due to work hardening phenomenon and applied plastic strain. Yang et al. [32] investigated the phase composition, mechanical behaviour, densification and microstructure of 316 L stainless steel by hybrid directed energy deposition and thermal milling process. It was found that the nearly fully dense 316 L steel specimens exhibit high microhardness under the optimum process parameters, which is attributed to the fine microstructure and the high density. Yang et al. [33] studied the critical maximum undeformed equivalent chip thickness for ductile–brittle transition of zirconia ceramics under different lubrication conditions. To explore the effect of lubrication conditions on the grinding behavior of zirconia ceramic, MoS_2_ nanoparticles with an average particle size of 50 nm were added into palm oil. It was found that the grinding behaviour of zirconia ceramics can be categorized into plastic removal, powder removal, elastic sliding friction and brittle removal. Cui et al. [34] proposed a new cryogenic nanolubricant minimum quantity lubrication (CNMQL) approach for grinding process that utilizes the heat transfer capacity of cryogenic air and antiwear/antifriction performance of nanolubricant. The authors developed the calculation formulas (defect ratio of workpiece surface and the energy ratio coefficient of the cooling medium) to evaluate grindability of Ti-6Al-4V titanium alloy. It was found that cryogenic nanolubricant shows significant improvement of convective heat transfer capacity. Wang et al. [35] investigated the cooling lubrication mechanism and technical iteration motivation of minimum quantity lubrication (MQL). The authors conducted comparative assessment of surface quality and tool wear under enhanced environmentally friendly lubrication turning, including parts enhanced by ultrasonic vibration, nanoparticles and textured tools. The development trends of MQL in turning operations for difficult-to-machine materials were fully reviewed. 

This work presents comprehensive SPIF tests of Al/Cu bimetallic sheets. Experiments were conducted on surface quality and drawpiece shape accuracy. Their main purpose was to assess the effect of vertical step depth Δz on the final properties of the drawpieces. Extensive research on the mechanical properties, surface roughness, hardness and structure was carried out. The temperature of the surface of formed sheets was measured during the SPIF process. The non-contact optical 3D scanning method was used to evaluate the geometry and sheet thickness of the samples after the SPIF process.

## 2. Materials and Methods

### 2.1. Test Material

The research material was Al/Cu bimetallic sheet (EN AW-1050A/Cu-M1E) in temper state z6 with a thickness of 1 mm. The ratio of Al/Cu layers was 1:1. The sheet was produced by the Roll Bonding (RB) method in industrial conditions in Walcownia Metali Dziedzice S.A. (Czechowice-Dziedzice, Poland). The chemical composition of the individual layers of Al/Cu bimetallic sheet determined using a Foundry-Master Xpert (Oxford Instruments Industrial Analysis, High Wycombe, UK) spectrometer is shown in Table 1 and Table 2.

The mechanical properties of the bimetallic sheet were determined in a static tensile test on a universal testing machine Instron 5566 (Instron Polska, Opole, Poland) according to the ISO 6892-1 standard. Tests were carried out at constant strain rate equal to 1 × 10^−3^ s^−1^. Test samples were cut at angles of 0°, 45° and 90° to the rolling direction of the sheet. Five samples were used for each cut direction. The average value of the measurements of the mechanical parameters of the sheet was calculated according to Equation (1).
(1)$$\bar{X}=\frac{X_{0}+2X_{45}+X_{90}}{4},$$
where X_0_, X_45_, X_90_—values of mechanical parameters for specimens cut at angles of 0°, 45° and 90°, respectively; $\bar{X}$—average value of the specific parameter.

### 2.2. Experimental Setup

The stretch-forming capacity of sheet metals was assessed using the Erichsen cupping test in accordance with the ISO 20482 standard. The test was carried out on a 142-40 universal cupping test machine (Erichsen Gmbh, Hemer, Germany) for testing sheets and strips, with a punch speed of 10 mm/min. The Erichsen index (IE) was determined based on the measurement of the depth of three bulges after the test was immediately stooped at the moment of fracture. Contact of the punch with both the Al and the Cu sides of the bimetallic sheet was considered. The surface roughness of the bimetallic sheet for Cu and Al side was measured with a T1000 Hommel Tester (JENOPTIK Optical Systems GmbH, Berlin, Germany). Measurements were conducted perpendicular to the sheet rolling direction. Five measurements were conducted and average values of mean roughness Ra were evaluated. To determine the hardness of the bimetallic sheets, a Tukon2500 tester with a Vickers pyramidal diamond tip was used; a load of 10 N was used.

The process of incremental sheet metal forming was performed on a CNC BF30 (Stürmer Maszyny Sp. z o.o., Kostrzyn, Poland) precision drilling–milling machine (Figure 1). The machine is numerically controlled by means of a computer station equipped with the Mach3 program and is also equipped with a dedicated table equipped with a sheet mounting system with two plates and screws. Dimensions of the square workspace were 100 × 100 mm.

The forming tool is composed of 145Cr6 hardened steel. The tip of the tool had a spherical end with diameter of 12 mm. Rapeseed oil was used as a lubricant. The oil flows continuously in the cavity of the drawpiece. This lubrication method is commonly used in SPIF [36,37]. Due to the concentrated character of the contact between the tool and the sheet in single point incremental forming, the phenomenon of friction between the workpiece and the tool plays an important role in the deformation of the material and the surface roughness of the finished components. Incorrectly selected lubrication conditions and forming parameters can lead to excessive temperature increase at the contact zone, thus reducing the effectiveness of lubrication. Materials susceptible to adhesive wear, i.e., aluminum and its alloys, require special attention in selection of forming parameters and lubrication conditions. The use of lubricants is essential at the interface between tool and workpiece providing tools with a longer life by improving heat distribution, reducing friction and wear, and removing waste materials [37,38]. Proper friction conditions are ensured mainly through the use of liquid and solid lubricants [39], optimization of the forming strategy [37], CNMQL [34] and MQL [35,40] approaches and modification of the tool surface [41]. Moreover, lubrication improves surface finish in SPIF [42]. Coolant oil has higher exergy than the grease as a consequence of difference using amount, which means more environmental burdens [39,43].

The tool path was square in shape, the dimensions of which decreased with the depth of a pyramid drawpiece (Figure 2a–c). Samples with dimensions of 125 × 125 × 1 mm were formed at a constant rotational speed of the punch of 600 rpm and with a constant table feed rate of 100 mm/min. The test conditions were established on the basis of a literature analysis [8]. The step size in the *Z* axis was in the range of 0.5–1.2 mm and was changed every 0.1 mm. This influenced the value of the wall angle and the height of the pyramid drawpiece. The angle of the pyramid drawpiece wall (Figure 2d) was determined on the basis of Equation (2). Depending on the unit step size Δz (with constant displacement in the sheet plane Δx = Δy = const.), the values of the drawpiece wall ranged between 45° and 68°. The height of the drawpiece was in the range between 50 (drawpiece no. 1 and 2) and 110 mm (drawpiece no. 13 and 14). Three samples were produced for each set of sheet incremental forming conditions. Table 3 shows the experimental design for SPIF of bimetallic sheets. 



(2)
$$\alpha =\mathrm{arctg}\left(\frac{\Delta z}{\Delta x}\right).$$



The 3D scans obtained with the Atos Core system were compared with a CAD model prepared using the GOM Inspect program. Measurements were obtained of the actual wall angles and their thicknesses. The results were compared with values calculated on the basis of the Equation (2) (wall angle) and Equation (3) (wall thickness):(3)$$t_{1}=t_{0}\cdot \sin\left(90-\alpha \right),$$
where t_1_ is the wall thickness of the drawpiece, t_0_ is the initial wall thickness, and α is the wall angle.

Based on the maps of geometric deviations of the scans from the nominal values, the maximum values (positive and negative) of surface deviations (max and min), average surface deviation (avg) and standard deviation of the surface (sigma) were obtained. Data were statistically interpreted within the three standard deviations (3σ).

The drawpieces no. 1 and 15 (Table 3) were subjected to microstructure examination and to static tensile tests using a Zwick Roell Z005 uniaxial tensile test machine. The samples intended for microstructure examination were obtained from the zone of undeformed sheet (Zone 1 in Figure 3a) and from the walls of the drawpieces: in the zone of the drawpiece formed perpendicular to the rolling direction of the sheet (Zone 2 in Figure 3a) and in the zone of the drawpiece formed parallel to the sheet rolling direction (Zone 3 in Figure 3a). Samples for the uniaxial tensile test were obtained in Zones 2 and 3 (Figure 3b). The thickness “t” of the samples depended on the wall thickness of the drawpieces.

Due to small bimetal sample sizes and unconventional geometry, it was necessary to use mini-test samples. The geometry (dimensions in millimeters) of the specimen is shown in Figure 3b. Samples were machined using a Mitsubishi wire electrical discharge machining (WEDM) machine which reduced the impact of additional stresses that may occur in the process. It is worth noting that the surface had some defects resulting from the SPIF process, although the effect is not too significant. To obtain statistically significant results, 4 samples were used in the experiment for each state.

### 2.3. Analysis of Variance

The purpose of the analysis of variance (ANOVA) is to determine the relationship between the input variables and the output variable using a mathematical equation of the appropriate order. This article uses a polynomial regression model. Step size and layer arrangement were included as input factors (Table 3). Two analyses of variance models were performed. In the first of these, the variable that was explained was the hardness of the sheet metal. In the second model, the output parameter was average roughness Ra. F statistics and backward elimination regression (BER) were used to determine the significance of the variables. The procedure of BER is explained in a previous paper by the authors [44]. In ANOVA, the variables were removed if the probability was not less than *p* = 0.1. A *p* value equal to 0.1 is required for backward elimination from the model [45]. The test of the significance of the regression model was performed by calculating the ratio of the mean square of the regression and the mean square of the error (F statistics) at the significance level α = 0.05. In the case of the ANOVA model for hardness, a negative predicted R^2^ = −1.02 was determined for the model with adjusted R^2^ = 0.9876. This implies that the overall mean may be a better predictor for hardness. Therefore, ANOVA results for hardness are not presented in this paper.

## 3. Results and Discussions

### 3.1. Mechanical Properties of as-Received Al/Cu Bimetallic Sheet

The stress–strain relationships for samples cut at an angle of 0°, 45° and 90° with respect to the sheet rolling direction are shown in Figure 4a. The values of the Rp0.2/Rm ratio, which describe the metal plasticity reserve, were also determined. The mechanical properties of the Al/Cu bimetallic sheet are anisotropic (Figure 4b). In the 0° and 90° directions, the yield stress and ultimate tensile stress have similar values and they are higher than in the 45° direction by approximately 7–8%. On the other hand, the greatest elongation occurs in the 45° orientation. The values of the Lankford coefficients, the mean normal anisotropy $\bar{r}$ and the planar anisotropy index Δr of the Al/Cu sheet are listed in Figure 4c. The tested material was characterized by a heterogeneous distribution of the Lankford coefficient in the sheet plane. For the orientation of 45°, significantly higher values of the r coefficient were found. On the other hand, the average value of the normal anisotropy coefficient was 0.53. This means that such material is susceptible to thinning during forming. The test sheet also exhibits a relatively high planar anisotropy index (Δr = −0.72). In the conventional deep drawing process of cylindrical drawpieces, a negative value of the high anisotropy index suggests the formation of ears at an angle of 45° with respect to the rolling direction of the sheet metal.

The average value of the IE index for a test sheet is approximately 7.4 mm. Contrary to [46], a higher IE value was found for the case in which the punch was in contact with the aluminium sheet layer during the test. There is an approximately 6% difference between the IE index value for cases in which the punch was in contact with the Al and that of the cases with the Cu sheet (Table 4). 

A similar dependence was also noted for the maximum force registered at the moment of appearance of the crack on the surface of the cup. In both cases, the surface of the cups was smooth and the shape of the crack was regular. Qualitatively, it can be stated that the test material was fine-grained and had a homogeneous structure.

### 3.2. Surface Quality

Table 5 shows the results of experimental tests on incremental forming of Al/Cu bimetallic sheet. Among the process conditions examined, only in the case of a step size of 1.2 mm (samples no. 15 and 16) did a crack appear on the wall during forming. Figure 5 shows selected samples after SPIF had been carried out. It was found that as the step size Δz increased, more distinct traces of the forming line appeared for both sheet layers. In the case of the outer surfaces of the walls of the drawpieces, no clear traces of the forming line were found. However, in the case of a step size of 1.2 mm, there are cracks near the corners of the formed drawpieces, both from the Al and the Cu side during forming.

### 3.3. Maximum Forming Angle

Figure 6a shows the method of determining the limit values of wall angles. The limit values of the drawpiece wall angle were determined. In the case in which the bimetallic sheet was formed on the Cu side, its value was 63.4°, while for the Al side it was 65.6° (the standard error of the drawpiece wall angle value was not greater than 0.06 degrees). Above these values, the drawpieces ruptured, and below these limit values, they were formed without loss of material continuity. Dependence of the side of the layer of the bimetallic sheet that was formed was also found. In the range of step sizes Δz examined, the actual mean wall angles, both in the samples formed on the Al and on the Cu sides, differed slightly from the values determined by Equation (2) (Figure 6b). The actual values of the wall angle after forming the sheets from the Cu side were more similar to those calculated from Equation (2) than in the case of the drawpieces formed from the Al side. The absolute mean deviations from the calculated values were 17.4′ for the Cu and 28.2′ for the Al side, respectively. Thus, it can be concluded that the tested material exhibits higher springback effect during forming on the aluminum side than on the copper side. It is assumed that the major source of springback in ISF process is global bending of sheet [47]. However, in the case of a bimetallic sheet, there is also an effect related to the stiffness and strength of the layers [48], hence the differences in the values of the tilt angle of the drawpiece wall depending on the sheet layer.

The deviation of the values of the drawpiece wall angle obtained by measurement from those that were calculated from Equation (2) was generally smaller for small step sizes. This is especially evident when comparing the wall angles of drawpieces no. 1 and 2 with drawpieces 13–16. The larger the step size, the more and more the formed angle deviates from the one that is calculated. With drawpieces formed with a step size of 0.5 mm, the values of the calculated and actual wall angles hardly differ. The greatest differences are visible for drawpieces formed with a step size of 1.2 mm. However, as shown in Figure 5, the samples broke prematurely. Additionally, one can see that the angle of each wall is different from the others (Figure 6). The angle of the wall opposite to the crack is the closest to the calculated value. Therefore, in the case of those samples that ruptured, it is necessary to take into account the possibility of disturbing the wall angle by breaking the continuity of the material.

### 3.4. Wall Thickness

The measurement of the thickness of the wall of the drawpieces was made on the basis of 3D scans obtained with the GOM Inspect software. The mean wall thicknesses of the drawpieces in the samples shaped on both the Al and Cu sides differed from the values calculated by Equation (3) (Figure 7). There is a noticeable difference in the case of samples formed on the Al layer side of the sheet at the following values of step sizes: 0.5 mm, 0.8 mm and 0.9 mm and above 1.1 mm. For these samples, the difference between the calculated and actual wall thickness was up to 10%.

With an increase in tool step size in the direction of the *Z* axis, the average thickness of the drawpiece wall decreased, which is consistent with other work [28]. At the same time, it was noticed that when forming the bimetallic sheet on the aluminum side of the sheet, greater thinning only occurred in the range of Δz values from 0.7 to 1.0 mm, which was confirmed by the results obtained in work [8]. However, in the step sizes in the ranges Δz ≤ 0.6 mm and Δz ≥ 1.1 mm, a greater reduction in the wall thickness was found when the bimetallic sheet was formed from the copper side of the sheet.

### 3.5. Deviation of Wall Geometry

By means of 3D scanning, deviations of the actual geometry from the CAD model were determined and were related to tool step size and the side of the material subjected to forming (Figure 8). The occurrence of wall springback effect can be observed from scans on which deviations from the nominal geometry are superimposed. This effect is visible from the smallest step size of 0.5 mm applied. The largest deviations from the expected shape of the drawpiece were found on drawpieces with a large wall angle formed at large values of step size. From the analysis presented, it can be concluded that the greater the wall angle α, the greater the springback effect. This effect can be minimised by using double sided Incremental Sheet Forming [49].

The surface deviations of the samples after forming had taken place on the aluminium and copper sides of the sheet are shown in Figure 9a,b, respectively. In step sizes Δz between 0.5 and 0.7 mm, the maximum deviations to the outside (max) and the minimum deviations to the inside (min) for drawpieces formed on both sides almost fully overlap and do not exceed the absolute value of 1 mm (Figure 9a). Visible deviation occurred in the case of the drawpiece formed on the copper side of the sheet with a step size of Δz = 0.8 mm (Figure 9b). The value of the maximum deviation is 2.27 mm (for a drawpiece formed on the aluminum side of the sheet: 0.86 mm), while the minimum deviation is −3.11 mm (for a drawpiece formed on the aluminum side of the sheet: −0.73 mm). The average values of deviations (avg) both for drawpieces formed on the aluminum side of the sheet and for drawpieces formed on the copper side of the sheet slightly differ from each other in the range of step size Δz between 0.6 and 0.8 mm (Figure 9b). For a step size of Δz = 0.5 mm, the average deviation is 0.6 mm, regardless of the side on which the bimetallic sheet is formed. Standard deviation in the surface (sigma) for step size Δz = 0.7 mm alone has an increased value compared to the other drawpieces (sigma = 0.6 mm), both for those formed on the aluminum side and copper side of the sheet. For a 1.2 mm step size, the drawpieces cracked during forming. As a result, when analysing the results of 3D scanning in the inspection program, the maximum and minimum deviations turned out to be much greater than those of the other step sizes.

### 3.6. Surface Roughness

Figure 10 shows the results of surface roughness measurements of sheet surface before and after the SPIF process. The roughness of the drawpieces formed on both sides of the bimetallic sheet increases with increase in step size Δz. The same conclusion was reached by Gheysarian and Honarpisheh [26]. Regardless of the impact side of the tool, the average roughness Ra values are very similar for the specific step size, which was also confirmed by the authors of the paper [27]. With the step size greater than 0.8, a greater difference in surface roughness was found. For Δz equal to 0.9 and higher, the copper surface was characterized by a higher Ra parameter.

The ANOVA results for the roughness of the sheet are shown in Table 6. The F value of 322.84 indicates that the model results are significant. There is only a 0.01% chance that such a large F value could be the result of noise. A *p* value for step size (A) below 0.05 indicates that this parameter is a significant factor in SPIF of the bimetallic sheet examined. The layer arrangement parameter (B) and its interaction with step size (A·B) have been eliminated from the model by backward elimination regression. The ANOVA equation in terms of the actual factors is as follows (Equation (4)):(4)$$\mathrm{Ra}=-4.90149+14.15476\cdot A-6.09524\cdot A^{2}.$$

The total correlation R^2^ of the regression model is 0.98 (Table 7). Due to the small difference between the adjusted R^2^ (0.9772) and the predicted R^2^ (0.9711), it can be concluded that the model is adequate. The calculated value of this coefficient in the model is over 42.6, so the regression model is adequate. Furthermore, the value of the signal-to-noise ratio parameter (adequacy precision) for a realistic model should be greater than 4. The value of 42.67 proves that the regression model can be used to navigate the design space.

A comparison of the experimental values of average roughness with the values predicted by the ANOVA model is presented in Figure 11a. The closer the points are arranged along a straight line, the better the prediction of the model is. Distribution of the residuals is proportional along the horizontal zero line (Figure 11b). It proves the normal distribution of the residuals in the model, which is required to verify the significance of the results obtained. The quadratic equation (Equation (4)) provides a very good approximation of the change in surface roughness with increasing step size (Figure 12a,b).

### 3.7. Hardness

The mean hardness of the material of the drawpieces subjected to incremental forming decreased in relation to the hardness of the as-received material (Figure 13). The hardness of the aluminum side of the bimetallic sheet was approximately 58 HV10 and of the copper side was approximately 118 HV10. In the step size between 0.5 and 1 mm, the hardness of the material of bimetallic drawpieces was greater on the copper side of the sheet, while for higher values of step size Δz = 1.1–1.2 mm, the relationship is reversed.

### 3.8. Microstructural Observation of CuAl Interface

The analysis of the interface was conducted on samples cut from the pyramids and the non-deformed fragments (Figure 14). Both materials seem to be closely attached to each other. No gaps or voids are seen in the micrographs of the edges of the sheet which demonstrate great adhesion between them. It is worth noting that there are some cracks on the outer surface which are a result of the tool runout.

### 3.9. Mechanical Properties of Bimetallic Sheet after SPIF

Figure 15 shows tensile curves of samples cut from drawpieces 1 and 15 on the basis of which their basic mechanical properties (Table 8) were determined. The main goal of the experiment was to analyse the change in mechanical properties and anisotropy of the bimetallic sheet material after forming. The tests were conducted for both Zones 2 and 3 (Figure 3). Overall, considering the dispersion of the results, there were no statistical results of the anisotropy of tensile strength. There were some more noticeable differences in the elongation, although they were the result of surface defects. Overall, it should be noted that the material was very homogeneous in terms of mechanical properties and microstructure. However, a decrease in sheet strength was found as a result of additive forming. The larger it is, the larger the step size used. The probable cause of this situation may be heating of the material during forming and an increase in grain size due to recovery and/or recrystallization [50]. This effect is also visible in the decrease in the hardness of the layers of the bimetallic sheet (Figure 11). The planned further research, including the analysis of the microstructure, should provide a 100% answer to the question about the cause of the phenomenon.

## 4. Discussion

The SPIF material forming of bimetallic sheets was the subject of analysis in the current study. Multiple methods were used to analyse the process parameters and the quality of the obtained product. The overall results and conclusions try to elaborate some critical data from the review concerning the formability of Cu–Al sheets. There was influence of the layer (from which the sheet was formed) on the deformability of the Cu/Al bimetallic sheet. For an angle of less than 55° and greater than 61°, better formability was achieved with forming from the copper side. This effect correlated well with the better stretch formability of sheets under biaxial tensile deformation (Figure 16). On the contrary, in the range from 55 to 66°, higher deformability was obtained for the aluminum layer.

In the current study, 3D scanning was used to obtain more in-depth analysis. The scanned geometry showed overall good correlation with theoretical equations. However, an issue arises with the springback effect which adds additional scatter of results. The overall surface roughness was in direct correlation with the ANOVA model which showed very good convergence with the current results. The increase in the roughness of the surface of the drawpieces, in relation to the sheet, on one hand, depends on the step size [26] and on the other hand is related to the increase in temperature during incremental forming [41]. Lastly, the mechanical properties and the microstructure of the bimetallic sheets were analysed, and it was found that copper and aluminum display very good adhesion which is additionally enhanced by interlocking of individual layers. The mechanical tests also showed a very uniform strength largely regardless of orientation which may be counterintuitive due to the heterogonous nature of the manufacturing method as copper and aluminum possess slightly different strengthening curve. Overall, it has been shown that the proposed methodology can be used to obtain a defect-free element. The key parameters in designing the process are step size and side of the bimetallic sheet. Based on the current practical results, it is possible to obtain a product with repeatable geometry and mechanical properties as well as microstructure.

## 5. Conclusions

The article presents the results of investigations on the effect of step size Δz of the forming tool on the formability and maximum forming angle, mechanical properties, hardness, surface roughness and microstructure of the Al/Cu bimetallic sheet drawpieces forming using SPIF. Two strategies of sheet formation were investigated, from the Al side and the Cu side of the sheet. Based on the research results, the following conclusions can be drawn:-The difference in the Erichsen Index IE value of the bimetallic sheet tested from the aluminum side of the sheet is 6% larger than that of the sheet formed from the copper side of the sheet;-Step size is the basic parameter that determines the formability of the material; increasing it to over 1.1 mm resulted in the rupture of the drawpiece material;-Step size also strongly influences the springback phenomenon; the larger the step size, the greater the deviation of the drawpiece geometry from the desired geometry;-The greater the step size, the more the wall angle after springback deviates from the desired profile;-Forming an Al/Cu bimetallic sheet from the aluminum side of the sheet permits drawpieces with a larger wall angle to be obtained;-The layer arrangement is not a statistically significant parameter influencing the average roughness of drawpieces;-From the two-layer arrangements examined in the range of step size between 0.7 and 1.0 mm, greater thinning of the sheet occurred during SPIF from the aluminium side of the sheet, while in SPIF of the bimetallic sheet from the copper side of the sheet, the greatest thinning was observed at Δz ≤ 0.6 mm and Δz ≥ 1.1 mm;-Observation of the sheet cross-section shows that no gaps or voids are seen in the surface joining the sheets, demonstrating great adhesion between the layers.

## Figures and Tables

**Figure 1 materials-16-00367-f001:**
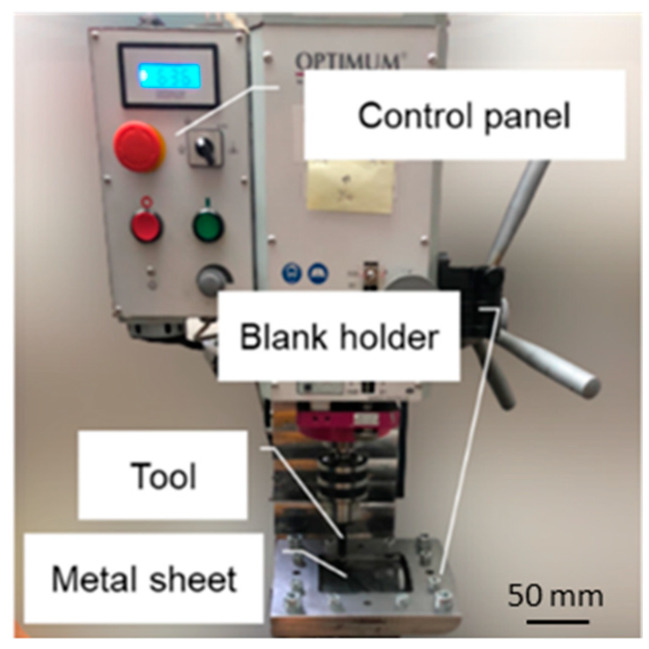
The CNC machine used for tests.

**Figure 2 materials-16-00367-f002:**
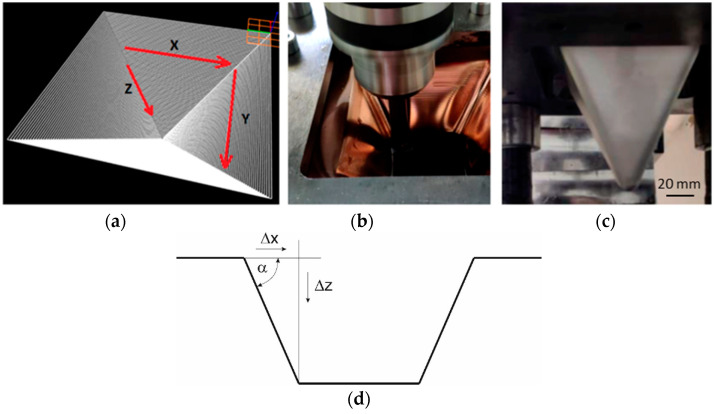
Trajectory of the forming tool (**a**), views of the internal (**b**) and external (**c**) surface of the pyramid drawpiece, scheme for determining the value of the wall angle α (**d**).

**Figure 3 materials-16-00367-f003:**
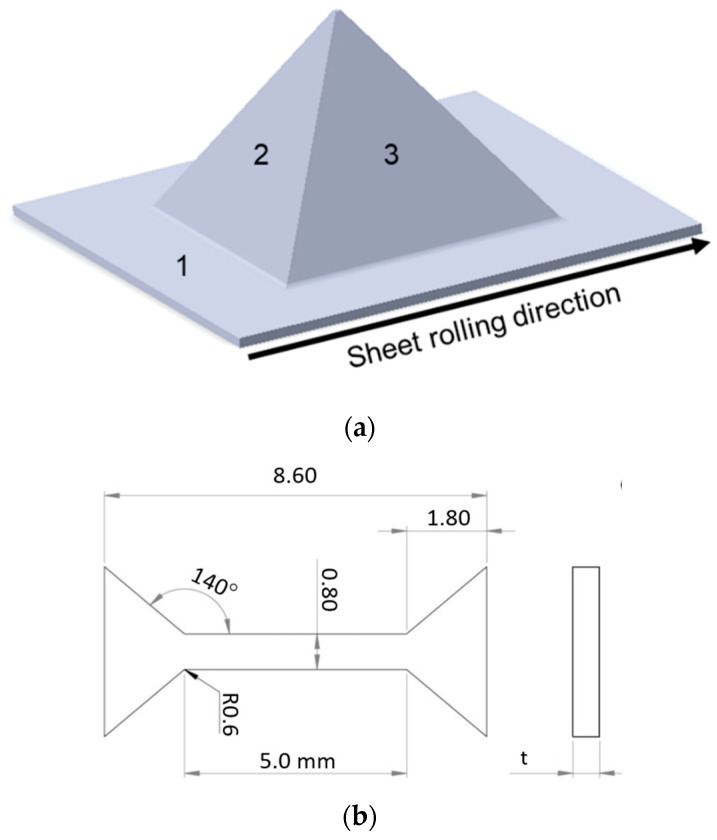
Location of excision of samples for the examination of microstructure and mechanical properties (**a**), mini-sample geometry used in the tensile tests (**b**).

**Figure 4 materials-16-00367-f004:**
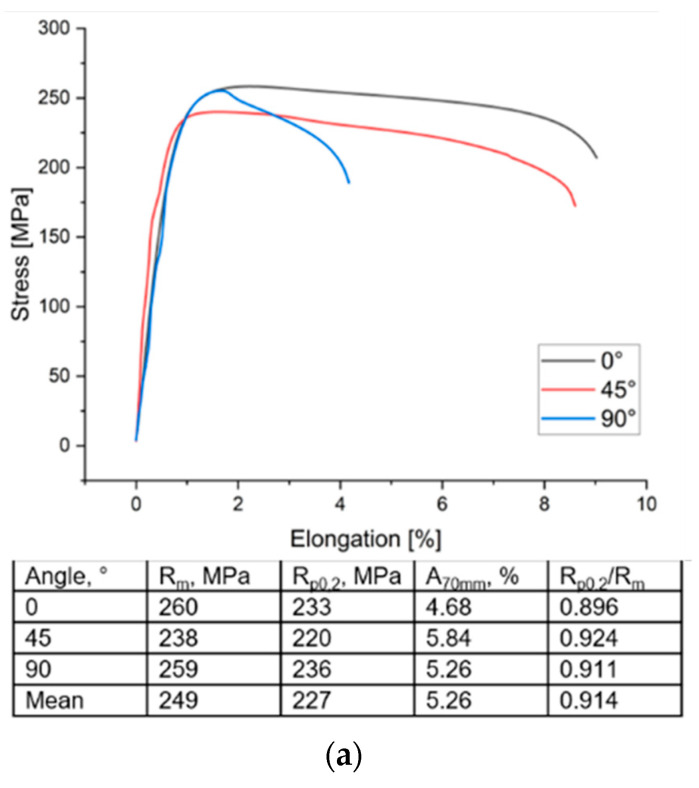
Stress–strain relation of the Al/Cu bimetallic sheet and basic mechanical properties of Al/Cu bimetallic sheet (**a**), influence of the sample cutting direction on the value of the yield strength R_p0.2_ and tensile strength R_m_ (units in MPa) (**b**) and distribution of the Lankford coefficient in the plane of the Al/Cu bimetallic sheet and plastic anisotropy parameters of Al/Cu bimetallic sheet (**c**).

**Figure 5 materials-16-00367-f005:**
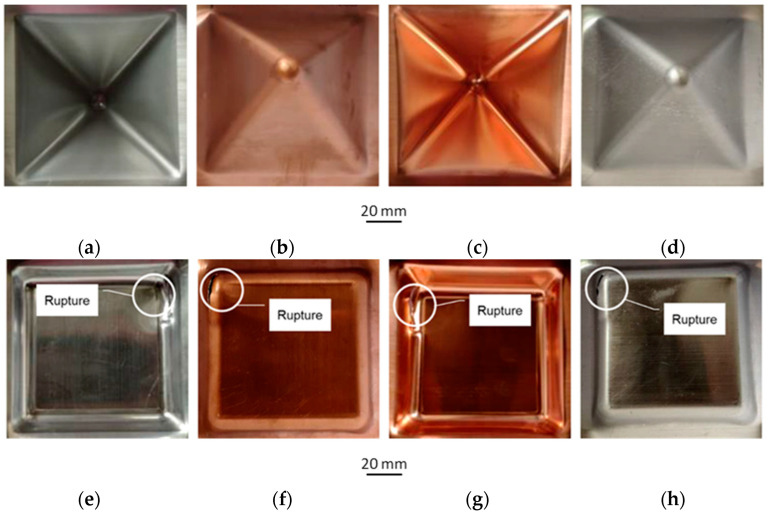
Photographs of drawpieces formed at Δz = 0.8 mm: SPIF on the Al side, sample no. 7 (**a**,**b**), sheet metal formed on the Cu side, sample no. 8 (**c**,**d**) and photographs of draw pieces formed at Δz = 1.2 mm: SPIF on the Al side, sample no. 15 (**e**,**f**), sheet metal formed on the Cu side, sample no. 16 (**g**,**h**).

**Figure 6 materials-16-00367-f006:**
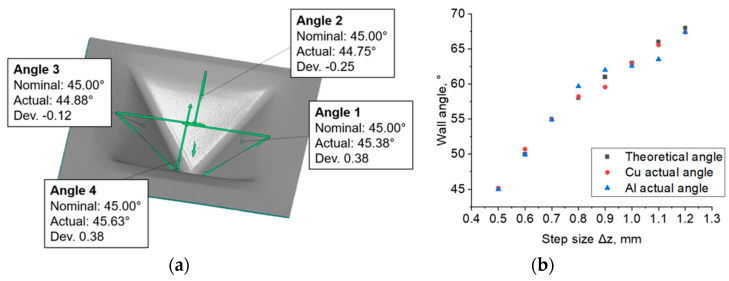
An example result of the wall angle measurement on the inside of the drawpieces for experiments no. 2 (**a**), variation of drawpiece wall angle depending on step size Δz (Δx = Δy = const. = 0.5 mm) (**b**).

**Figure 7 materials-16-00367-f007:**
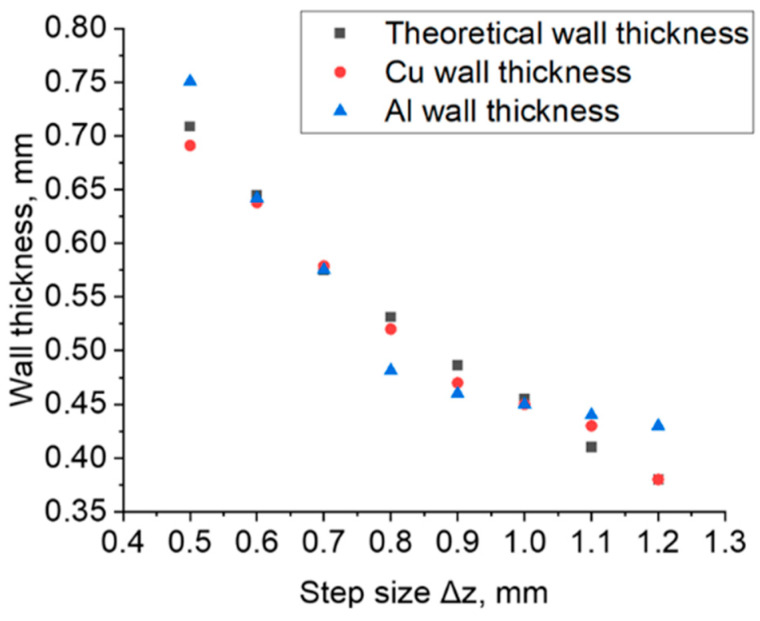
Variation in the average wall thickness of the drawpiece depending on step size Δz (Δx = Δy = 0.5 mm).

**Figure 8 materials-16-00367-f008:**
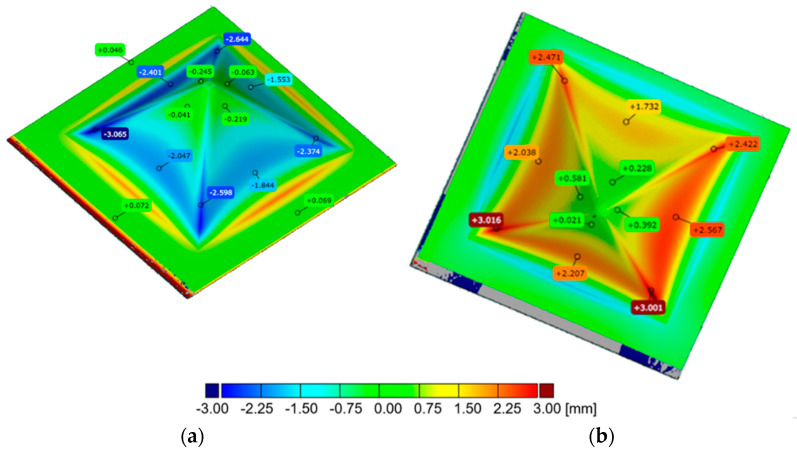
Distribution of the deviations of the drawpiece geometry: (**a**) drawpiece no. 1 (Δz = 0.5 mm, h = 40 mm) formed on Al side of the sheet, (**b**) drawpiece no. 2 (Δz = 0.5 mm, h = 40 mm) formed on Cu side of the sheet, (**c**) drawpiece no. 13 (Δz = 1.1 mm, h = 40 mm) formed on Al side of the sheet, (**d**) drawpiece no. 14 (Δz = 1.1 mm, h = 40 mm) formed on Cu side of the sheet.

**Figure 9 materials-16-00367-f009:**
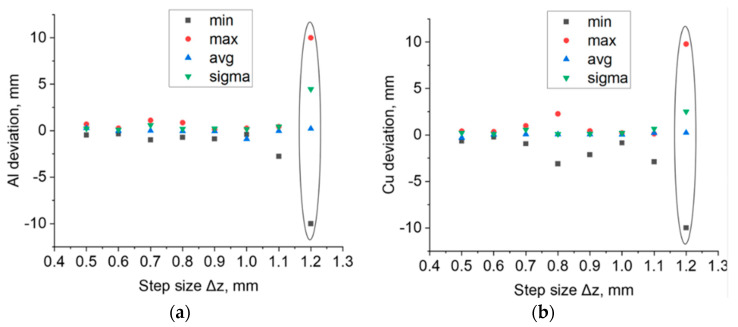
Maximum positive surface deviation, maximum negative surface deviation, average surface deviation and standard deviation for drawpieces formed on the (**a**) aluminum side of the sheet and (**b**) copper side of the sheet (**b**). The graphs also show the results for the samples that cracked.

**Figure 10 materials-16-00367-f010:**
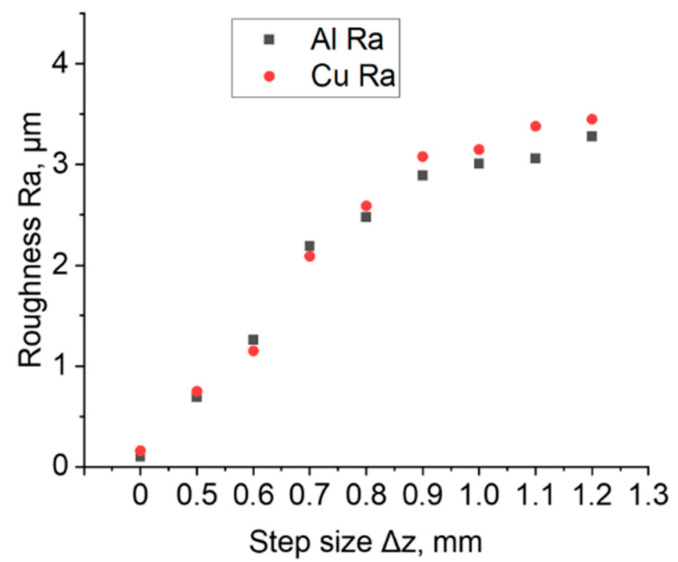
Average roughness Ra of the samples in the initial state (Δz = 0 mm) and of samples formed on both sides of an Al/Cu bimetallic sheet.

**Figure 11 materials-16-00367-f011:**
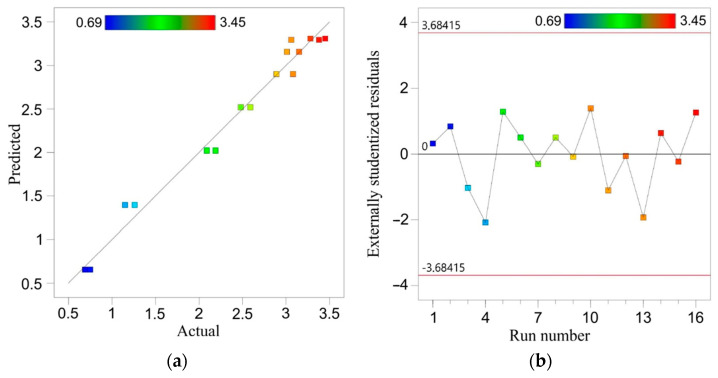
Predicted vs. actual values of average roughness Ra (**a**) and distribution of the externally studentized residuals through the run number (**b**).

**Figure 12 materials-16-00367-f012:**
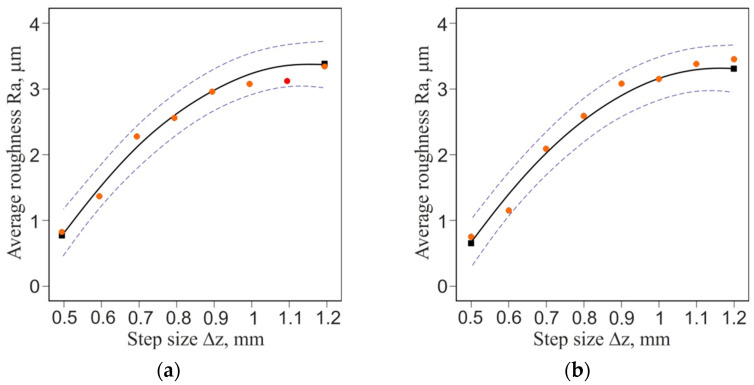
One-factor graph for the average roughness Ra parameter for formed side of bimetallic sheet: (**a**) Al and (**b**) Cu.

**Figure 13 materials-16-00367-f013:**
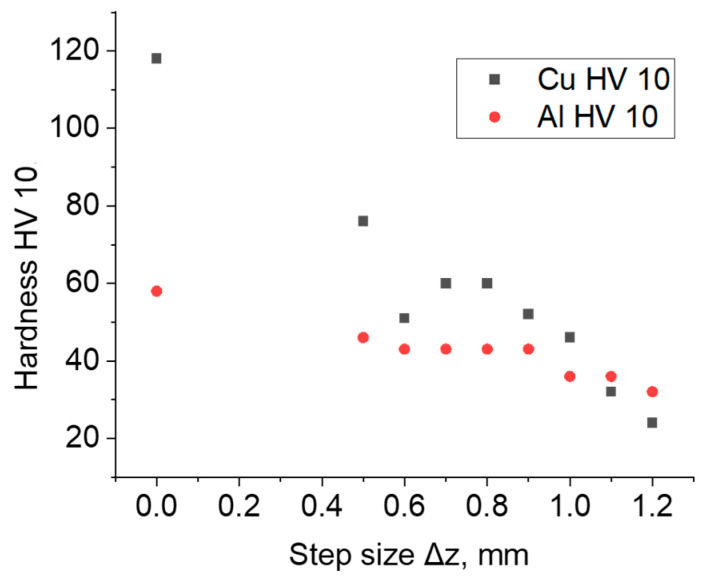
Hardness HV10 of the samples in the initial state and formed on both sides of Al/Cu bimetallic sheet.

**Figure 14 materials-16-00367-f014:**
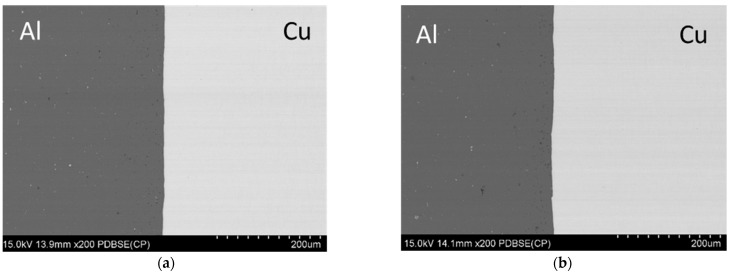
The microstructure of the material in the as-received state (**a**), the microstructure of the material deformed in the direction parallel (**b**) and perpendicular (**c**) to the sheet rolling direction (Δz = 0.5) and the microstructure of the material deformed in the direction parallel (**d**) and perpendicular (**e**) to the sheet rolling direction (Δz = 1.2).

**Figure 15 materials-16-00367-f015:**
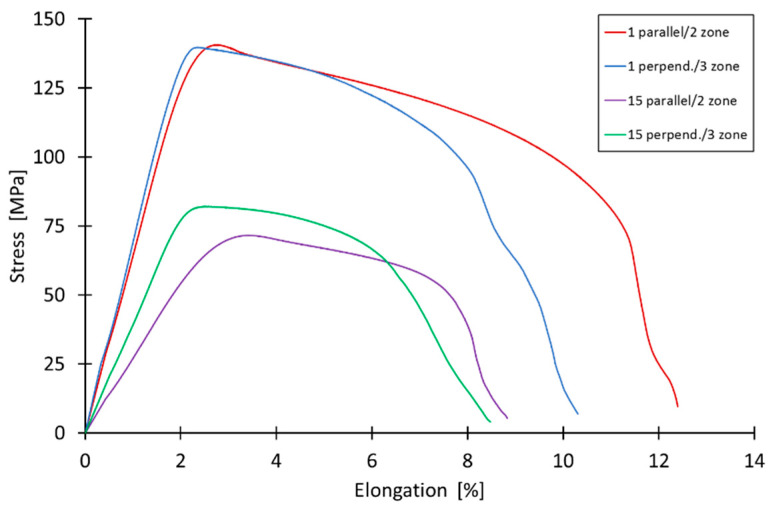
Tensile curves of as-received bimetallic sheet and of specimens cut from drawpieces no. 1 and 15.

**Figure 16 materials-16-00367-f016:**
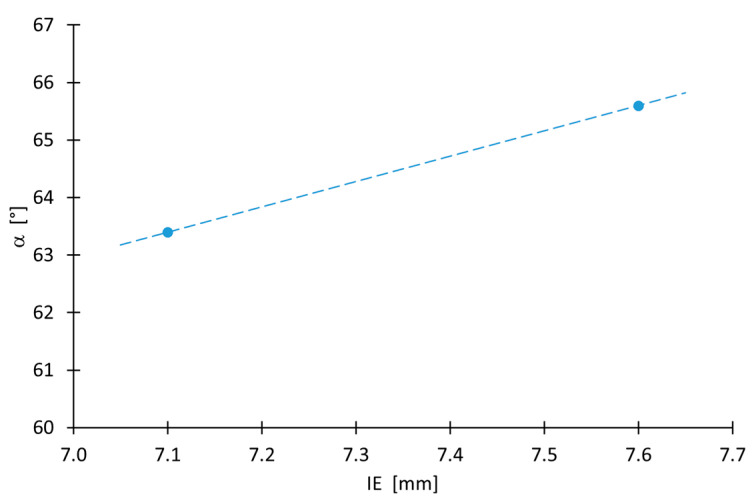
Relationship between the tilt angle of the drawpiece wall and the value of the Erichsen index for α_teor_ = 65.

**Table 1 materials-16-00367-t001:** Chemical composition of technically pure aluminum EN AW-1050A (wt. %).

Al	Mg	Mn	Fe	Si	Cu	Zn	Ti	Bal.
99.43	0.025	0.027	0.23	0.17	0.028	0.033	0.029	0.028

**Table 2 materials-16-00367-t002:** Chemical composition of Cu-M1E (wt. %).

Cu	Bi	O	Pb	Bal.
99.93	0.0004	0.036	0.004	0.029

**Table 3 materials-16-00367-t003:** The experimental design for SPIF of bimetallic sheets.

Drawpiece No.	Step Size Δz, [mm]	Wall Angle, [°]	Layer Arrangement	Tool Rotational Speed, [RPM]	Table Feed Rate, [mm/min]
1	0.5	45.0	Al	600	100
2	0.5	45.0	Cu		
3	0.6	50.2	Al		
4	0.6	50.2	Cu		
5	0.7	54.5	Al		
6	0.7	54.5	Cu		
7	0.8	58.0	Al		
8	0.8	58.0	Cu		
9	0.9	61.0	Al		
10	0.9	61.0	Cu		
11	1.0	63.4	Al		
12	1.0	63.4	Cu		
13	1.1	65.6	Al		
14	1.1	65.6	Cu		
15	1.2	67.4	Al		
16	1.2	67.4	Cu		

**Table 4 materials-16-00367-t004:** The results of the stretch-forming capacity of Al/Cu sheets.

Contact Layer	Al	Cu	Average Value
IE, mm	7.6	7.1	7.4
F_max_, kN	8.5	8.0	8.3
Photo of the samples	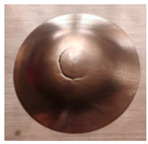	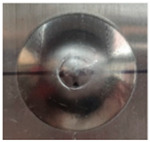	
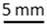

**Table 5 materials-16-00367-t005:** The quality of the drawpieces after visual assessment (Y—meets the requirements, N—does not meet the requirements).

Trial No	Visual Quality Requirements			Comments
	Regular Shape	Smooth Surface	End of Path	
1–13	Y	Y	Y	-
14	N	Y	N	The sample ruptured at a tool depth of 5 mm
15	N	Y	N	The sample ruptured at a tool depth of 10 mm
16	N	Y	N	The sample ruptured at a tool depth of 15 mm

**Table 6 materials-16-00367-t006:** Results of the ANOVA for the regression model of average roughness Ra.

Source	Sum of Squares	Degree of Freedom	Mean Square	F Value	*P* Value	Meaning
Model	13.33	2	6.67	322.84	<0.0001	significant
A–step size	12.08	1	12.08	585.23	<0.0001	
A^2^	1.25	1	1.25	60.46	<0.0001	
Residual	0.2684	13	0.0206			
Correlation Total	13.6	15				

**Table 7 materials-16-00367-t007:** Fit statistics of the regression model for average roughness Ra.

Standard Deviation	0.1437
Mean	2.41
Coefficient of variance, %	5.97
R^2^	0.9803
Adjusted R^2^	0.9772
Predicted R^2^	0.9711
Adequacy precision	42.67

**Table 8 materials-16-00367-t008:** Basic mechanical parameters of as-received material and of specimens cut from drawpieces no. 1 and 15.

Sample	Yield Stress, [MPa]	Max. Elongation, [%]	Ultimate Tensile Strength, [MPa]
As-received material (Figure 7a)	227	5.3	249
1 parallel	137	11.5	141
1 perpendicular	138	10.3	140
15 parallel	66	8.2	72

## Data Availability

Data are available with the first author and can be shared with anyone upon reasonable request.

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
