# Peer review of "Effect of Step Size on the Formability of Al/Cu Bimetallic Sheets in Single Point Incremental Sheet Forming"

_materials, 2022, doi:10.3390/ma16010367_

Round 1

Reviewer 1 Report

The authors present the results of single point incremental sheet forming in Al/Cu sheets. The objective is to analyse the influence of the step size in some mechanical properties, and in surface roughness and shape accuracy.

Although this kind of studies have already been accomplished by other authors in Al/Cu sheets, they have been performed in Al/Cu sheets with different Al and Cu thickness ratios. The results are interesting, but some improvements are needed before publishing:

1.   The following article presents a similar study in Cu-Al sheets but with aluminium and copper layers thicknesses of 0.85 mm and 0.15 mm, respectively, and also with 0.9 mm and 0.1 mm.

“Single point incremental forming of Cu-Al composite sheets: A comprehensive study on deformation behaviors” Archives of civil and mechanical engineering 19 (2019) 484-502 https://doi.org/10.1016/j.acme.2018.11.011

This paper should be included in the references and the current results comparing with those found in that paper.

Section 2.2

2.      Please, specify you are going to make the surface roughness and hardness measurements from both, the Al and the Cu side.

3.      I do not understand well the test conditions and the different formed samples you got. The starting materials were sheets with 125 x 125 x 1 mm in size. The step size in z axis was in the range 0.5-1.2 mm, but which was the final depth of the imprints?  I guess you want to investigate the effect of getting the same  final geometry (same final depth of the imprints) but with a different z-step size, which means that with smaller steps more “hits” are necessary.

It is written that the step size was changed every 0.1 mm, but it means that the imprints are overlapped, as the x-step size are 0.5 mm or low. It is very confusing.

4.      Figure 6 illustrates the location of the samples for microstructure and mechanical properties. For that, individual imprints need to be made. Please clarify taking into account the comments of the previous paragraph. Figure 6 b shows the dimensions and geometry of the samples used for the tensile tests. Are they given in millimetres? Is 0.6 mm the thickness of the tensile samples? What is the final depth of the pyramid?, Note that the length of the samples for tensile tests are 8.6 mm.

Section 2.3

5.      Line 218. There is a mistake. Table 2 should be replaced by table 3.

Section 3.1

6.      Explain in line 234 what is Rp0.2 and Rm. Also explains what is the meaning of A70 mm in figure 7

7.      Are the Lanford coefficients obtained at a specific stress value?

Section 3.3

8.      Please note that the comments in table 7 indicate different tool depths (what I suppose I have called final depth of the imprints). I understand to get different tool depths when you are studying the resistance to rupture but not when you are studying the surface roughness.

9.      Please, include in the captions of figures 11 and 12 the number of the corresponding samples.

Section 3.4, 3.5 and 3.6

10.   I suggest and simplify these sections, as I feel like some information is redundant.

11.   Tables 8 and 9 should be unified and compacted.

Section 3.8

12.   In which parts of the pyramids have been performed the hardness measurements? There is anisotropy.

Section 3.9

13.   The microstructure study is very poor. You have only demonstrated the adherence, but this is not a microstructure analysis, as with these micrographs the grains are not visible. It would be adequate to analyse more in depth the microstructure, studying the grain morphology.

Section 3.10

14.   How you compare the tensile curves of figure 22 with those obtained from figure 7?

Section 3.11

15.   I find the sentence from line 470 to 472 confusing.

Reviewer 2 Report

The paper reports the effect of step size on the formability of Al/Cu bimetallic sheets 2 in single point incremental sheet forming. The topic is quite interesting and I think beneficial for the industry. The characterization presented in the paper is comprehensive. The figures are attractive. All the results is discussed in a good way that referred to other scientific reports. I think this is a good paper. One improvement that I suggest is the abstract should include quantitative data of the results.

The main question addressed by the research is the effect of step size of a forming tool on the characteristics of Al/Cu bimetallic sheet. In the Introduction, authors mention that the ISF method has been patented. In the present work, does the step size is modified based on the patent? Why is step size important? The methodology used in the paper is quite good. But how does the author determine to use 0.1 mm step difference in the delta z? The font in the image 6b and 7 is too small. I suggest enlarging the font.  The calculation of wall angle in Table 8 and Fig. 14, how many samples do the author use? Does the result reproducible? Author shall describe the fractography pattern in Fig. 23 before end up with the claim that Cu/Al sheet has good adhesion. Does the author measure the adhesion quantitatively?

Reviewer 3 Report

The novelty should be clearly presented in the abstract. One of the weaknesses of the abstract is the quantitative results and data that should be added to it. The abstract is mostly used to present the research method and it is better to improve it by presenting innovation in a transparent manner and quantitative results.

The number of Figures and tables provided is very large. Some of these images can be deleted or merged together.

The introduction is written very superficially. It is suggested to use new and strong articles in this field.

https://doi.org/10.1080/02670836.2020.1867784

https://doi.org/10.1016/j.jmatprotec.2020.117023

https://doi.org/10.1016/j.ijmachtools.2017.06.003

https://iopscience.iop.org/article/10.1088/2053-1591/ab6408

https://doi.org/10.1016/j.jmapro.2022.06.003

https://doi.org/10.1016/j.triboint.2022.107766

Figures 1, 4 and 5 can be deleted.

Tables 4 and 5 should also be deleted. The data in these tables have no research value.

Figure 3 should be merged with Figure 2.

Discuss the role of the lubricant in the investigated parameters. Also, the method of lubrication operation should be provided. Use the resources suggested in this section.

Merge Figures 7 and 8 together. Also, the results presented in the figures are of poor quality. Improve the quality of these figures.

Figures 9 and 10 do not provide good information about the heat distribution during the forming operation and different areas of the sample. Use better images or remove this section.

Figures 11 and 12 should be merged together. This section only reports the results. The results should be compared and discussed.

This issue can be seen in almost all parts of the article.

In the Fractography section, the results do not match the figures. The authors claim that soft fracture is observed along with the dimple, while the images show the opposite (brittle), which is due to oxidation of the sample surface. These images can be deleted along with this section. Also, to show the quality of adhesion between the layers, it is better to use the peeling test.

Round 2

Reviewer 1 Report

The authors have considerably improved the writing of the manuscript, but there are still some points and amendments that need to be clarified and improved.

As a general comment, many figures of the original manuscript have been removed (I agree with that) but those of the new version have not been renumbered.

Abstract

                The content of the abstract has been improved by presenting the quantitative results of this work but it is too long. The description of the research method should be compacted and some parts deleted.   

Section 3.3

                It has been included both in the text (lines 348-351) and in the figure caption the enumeration of the samples depicted in the new figure (old 11 and 12 figures). Please remove it from the text, since that makes it difficult to read.

Section 3.9

                The tittle of this section should be changed to another more specific to the subject it deals with, which is only the analysis of the interface.

                To better appreciate the morphology of the interface micrographs with a higher magnification should be included.

                Please, the scale bars are very hard to see.

Section 3.10

                New paragraphs have been included to discuss the results. Please revise the writing of some sentences.

Reviewer 3 Report

Almost all the referee's comments have been answered. It is only suggested to add a scale bar in all used images.
